# Research on Green Technology Path of Cold-Chain Distribution of Fresh Products Based on Sustainable Development Goals

**Lei Zhou, Qianpeng Li \*, Fachao Li and Chenxia Jin**

School of Economics and Management, Hebei University of Science & Technology,
Shijiazhuang 050018, China
* Correspondence: liqianpeng15@126.com

**Abstract:** In order to meet customers' demand for freshness and the time window of fresh product distribution, and achieve the sustainable development of the fresh product cold-chain logistics industry, green technology is used to solve the optimization problem of the cold-chain distribution path of a variety of fresh products, taking transportation, refrigeration, and carbon emission cost as the objective functions. A hybrid particle swarm optimization (HPSO) algorithm was designed to solve this problem with the minimum freshness requirement of different kinds of cold-chain distribution. The results show that when the minimum freshness requirement of each fresh product is 75%, the total distribution cost obtained by the hybrid particle swarm optimization algorithm is CNY 4218, and the customer satisfaction is 88.22%. While satisfying the freshness constraint, the results obtained by the particle swarm optimization algorithm reduce the total distribution cost by CNY 362 and improve customer satisfaction by 3.82%. Green technology is beneficial to reduce vehicle pollution emissions and the loss of fresh product resources, thus achieving sustainable development.

**Keywords:** green technology; path planning; cold-chain logistics; total distribution cost; hybrid particle swarm optimization; multi objective optimization

## 1. Introduction

In recent years, with the popularity of the Internet, the development of fresh e-commerce has entered a new stage. The demand for fresh products is gradually increasing, and the price and quality of fresh products are increasingly related to people's daily life. At present, in the process of transporting fresh products from distribution centers to customers through cold-chain logistics, there exist several problems faced by Chinese fresh e-commerce enterprises, such as high total distribution costs, the low freshness of products, and large distribution pollution emissions. In view of the fact that the perishability of fresh products causes resource consumption and high timeliness requirements, fresh distribution enterprises are required to meet higher requirements than traditional logistics distributors. Fresh products that fail to meet the customer's requirements when delivered are usually rejected. Whether the delivery is delivered on time is closely related to customer satisfaction. Therefore, we introduce the development status of green logistics, and conduct research on the vehicle routing problem (VRP) of multiple types of fresh products. Through green technology, explore ways to reduce consumption of fresh products and vehicle pollution emissions to achieve the sustainable development of the cold-chain logistics industry.

Green logistics refers to the process of using advanced logistics technology to reduce the impact of logistics on the environment. With the development of the economy and the

aggravation of environmental pollution, green logistics is more important for achieving sustainable development, and many research achievements have emerged at home and abroad [1–5]. Agyabeng et al. [6] used the partial least squares structural equation model to analyze the relationship between green human capital and green logistics practice, green competitiveness, social performance, and financial performance. Kwak et al. [7] studied the factors of participants' willingness to use the green logistics platform. Through structural equation model analysis, the research shows that perceived usefulness has a greater impact on participants' intention to use green logistics platforms. Wu [8] conducted research on how to build green industry and strengthen green logistics management in a low-carbon economic environment. Tan et al. [9] proposed a green logistics framework based on blockchain, integrating the Internet of Things and big data and improving the competitiveness of logistics enterprises. Zhang et al. [10], based on the perspective of green environmental protection, used the ant colony algorithm to build the ghost fox model of a distribution-vehicle path so as to shorten the distribution distance and reduce vehicle pollution emissions.

VRP is a classical non-deterministic polynomial problem that has been studied by many experts and scholars in China and abroad [11–14]. In terms of customer satisfaction with fresh distribution, Wang et al. [15] proposed the GA-TS algorithm to plan the logistics distribution of fresh products from two aspects: time window and temperature control. The advantage is that the proposed genetic algorithm achieves the optimal planning of the logistics distribution of fresh products, while other factors affecting the logistics distribution were not fully considered in their study. Qin et al. [16] established an optimization model of the integrated cold-chain vehicle routing problem with the minimization of customer satisfaction cost as the objective function. Considering the combination of heterogeneous vehicles, the advantage of their study was that the established optimization model could improve the efficiency of the vehicle path distribution, but the angle was single, and their study did not consider the influence of other factors on the vehicle path. Gu and Yang [17] designed a genetic algorithm for the delivery of frozen food for customers, and drew a conclusion by comparing the results of different methods. The advantage of their study was that the designed genetic algorithm provided support for improving the customer efficiency of frozen food delivery, but their algorithm could not fully consider the influence of other factors. The objective function of Zhao et al. [18] is to minimize the total economic cost and new loss costesearched the electric vehicle routing problem considering traffic networks and customer time-window requirements, and designed an improved ant colony algorithm to solve the problem. The advantage of their study was that the improved ant colony algorithm could improve the freshness of the product; however, the disadvantage was that influencing factors outside the traffic network and the customer time-window needs were not fully considered. Ji et al. [19] constructed the cold-chain fruit distribution mode with efficiency and satisfaction as the dual indexes, and used the optimized genetic algorithm to solve this mode and compared the results with the above methods.

In terms of the freshness of fresh products, the advantage of this study is that using the optimized genetic algorithm can greatly improve efficiency, while the disadvantage is that the influence of other regional factors on the product freshness is not fully considered. Liu et al. [20] established a dual-objective design mode that considered carbon emission cost, realized the freshness target provided by users for products, and conducted a solution analysis. Their design mode had a certain effect on the improvement of product freshness, but the disadvantage was that the influence of other factors on product efficiency was not fully considered. Liu et al. [21] designed an improved genetic algorithm solution model based on the tabu search according to the characteristics of fresh agricultural products' quality assurance and low-carbon green logistics, and considered the decrease of freshness to penalize production costs and carbon emission costs. Their study fully considered the quality of agricultural products and the characteristics of low-carbon green logistics, but the disadvantage was that they did not fully consider the freshness of the

products. Zhao [22] designed an improved ant colony algorithm to solve the multi-objective optimization model based on cost, carbon emission, and customer satisfaction. The advantage is that the designed algorithm guarantees the quality of the product, while the disadvantage is that there lacks considerations of the influence of other factors on the freshness of the product. Jiu et al. [23] constructed the freshness function of fresh products, designed the algorithm to solve it and verified the rationality of the model and the effectiveness of the algorithm. The advantage of their research is that the freshness of the product is fully considered, while the disadvantage is that the influence of other factors on the product distribution efficiency is not fully considered.

Based on the above analysis and the summary of domestic and foreign scholars' discussion on the VRP problem, there are few studies on the green technology vehicle routing planning under freshness constraints for a variety of fresh products, and there is a lack of articles on the impact of freshness constraint and change on the total distribution cost and customer satisfaction. In order to maximize customer satisfaction and minimize the total cost of fresh distribution, this paper divides the total cost into fixed, transportation, refrigeration, and carbon emission costs and establishes a multi-objective vehicle routing planning model with time window and perishability. Each customer has a fuzzy time window requirement for the delivery of fresh products and a service time requirement for the loading, unloading, and handling of fresh products. The freshness model of fresh products is used to evaluate the quality of fresh products so as to meet the customer's requirements for fresh products. In the process of solving the model, the hybrid particle swarm optimization algorithm is designed by combining the particle swarm optimization algorithm with the genetic algorithm. By analyzing the constraints of freshness and the minimum constraints of changing the freshness, the example is verified. The effect of freshness on customer satisfaction and the total cost is analyzed, and the effectiveness of the hybrid particle swarm optimization algorithm is verified by a comparison with other intelligent optimization algorithms.

## 2. Model Building

### 2.1. Problem Description

This paper studies the Multi-Objective Vehicle Routing Problem with Time Windows (MOVRPTW) and constructs the dual Objective function of the lowest total delivery cost and the highest customer satisfaction. Factors restricting the freshness of multiple types of fresh products are also considered.

The distribution center can satisfy the demand for fresh commodities at each demand point and includes multiple refrigerated trucks. The cargo carrying capacity and the cost of each refrigerated truck are consistent. The truck travels from the distribution center to distribute multiple fresh products to demand points and returns to the distribution center after the transportation task is completed. The geographical location and demand of each demand point are known. The refrigerated vehicle has the maximum load limit, and it is required to deliver fresh products to the demand point within a specific time period. The optimization objective can be achieved through a reasonable driving route, which can reduce the total cost of distribution and increase customer satisfaction. Results are analyzed to find out whether there are freshness constraints, to verify the rationality of freshness constraints, and then to obtain the optimization results of the hybrid particle swarm optimization algorithm, particle swarm optimization algorithm, and genetic algorithm and to analyze them for the verification of the effectiveness of the model and the algorithm.

*2.2. Modeling Ideas*

The objective function is to minimize the total cost of cold-chain logistics of fresh goods, to maximize customer satisfaction, and to comprehensively consider the freshness constraint and the fuzzy time window of fresh products. In order to solve the problem from the perspective of theory and feasibility [24]. The established multi-objective vehicle route planning model mainly considers the following aspects.

(1) Reasonably plan the path to prevent delivery time delay or vehicle overload caused by improper transportation of vehicles.

(2) Considering the minimum total cost of distribution, under reasonable premises and constraint conditions, satisfy the requirements of all customers' time windows in the shortest possible distance and in the shortest possible time.

(3) Customer satisfaction is reflected through the delivery time of fresh products to meet the fuzzy time window requirements of different customers. The loading and unloading service time of each customer is also considered.

(4) Customer demand points have high requirements for the freshness of fresh products, so the freshness of fresh products is an important part of the cold-chain distribution process. The model adds the freshness constraint, considering the freshness constraint of different kinds of fresh products.

*2.3. Model Hypothesis*

In this paper, the cold-chain distribution path planning model of fresh products needs to be carried out under specific conditions. There are many uncertain factors in the actual cold-chain distribution of fresh products, such as traffic congestion, weather conditions, and other random factors. These factors are unavoidable in the real distribution of fresh products, which will cause some errors in the final results of the model calculation, but fortunately, they will not affect the rationality of the experimental result planning in this paper. The following assumptions are made in this paper:

(1) Assuming that the distribution route of cold-chain fresh products is unobserved, and the speed of refrigerated trucks in fresh distribution is regarded as a uniform movement;

(2) The total load weight and the route planning scheme of refrigerated trucks can meet the total demand of customers at each demand point;

(3) The delivery center can meet the demand of all customers for fresh products, and the vehicle can provide service to any demand point;

(4) The time window, demand, location requirements, and other parameters of customers at each demand point are known and can be served by one vehicle;

(5) The refrigerated vehicles have the same model parameters and are in good condition;

(6) The starting point and the endpoint of the refrigerated truck's fresh product distribution route are the distribution center;

(7) The refrigerated truck has a constant internal and external temperature during the distribution of fresh products in the cold chain.

*2.4. Symbol Description*

(1) Collection: *N* represents the collection of fresh customer demand points $\{n \mid n = 1,2,i\ldots,j\ldots,N\}$; $i = 0$ represents distribution center *O*. *p* represents the collection of fresh product types, $\{p \mid p = 1,2,3,\ldots P\}$; *K* represents the collection of vehicles, $\{k \mid k = 1,2,3\ldots,K\}$.

(2) Parameter: $F_k$ indicates the inherent cost of the vehicle; $Q_k$ indicates the maximum carrying capacity of the vehicle; $q_{ip}$ indicates the quantity of fresh products *p* ordered at the demand point *i*; $l_p$ indicates the minimum freshness requirements of fresh products *p*; $\Omega_p(t_i)$ indicates the freshness of fresh products *p* after reaching the demand

point; $T_p$ indicates the effective life cycle of fresh product *p*; $d_{ij}$ shows the distance between *i* and *j*; $V$ indicates the average vehicle speed; $f_c$ indicates the unit cost of carbon emissions; $C_k$ indicates the cost per unit distance of vehicle use; $\alpha$ represents the unit price of refrigerant; $\beta$ represents the refrigerant consumption per unit time; $t_{ij}$ represents the transportation time from point *i* to point *j*; $[t_{i2}, t_{i3}]$ indicates the satisfactory time window of demand point *i*, $t_{i2}$ is the upper limit and $t_{i3}$ is the lower limit; $[t_{i1}, t_{i4}]$ indicates the acceptable time window of demand point *i*, $t_{i1}$ is the upper limit; and $t_{i4}$ is the lower limit.

(3) Decision variable: $X_{ojk}$ indicates that the refrigerated vehicle *k* transported from the distribution center to the demand point *i* is 1, or 0. $X_{ijk}$ indicates that the refrigerated vehicle *k* transported from the demand point *i* to the demand point *j* is 1, or 0; $Q_{ijk}$ indicates that the refrigerated vehicle *k* transported from the demand point *i* to the demand point *j* is 1, or 0.

*2.5. Calculation of Relevant Indicators*

2.5.1. Customer Satisfaction

In Equation (1), $[t_{i2}, t_{i3}]$ is the changing time window of customer satisfaction [2]. In this period, customer satisfaction is 1. When the delivery time is within the time window $[t_{i1}, t_{i2}]$, customer satisfaction is proportional to time. When the delivery time is within the time window $[t_{i3}, t_{i4}]$, customer satisfaction is inversely proportional to the time. When the delivery time is outside the customer's acceptable time window $[t_{i1}, t_{i4}]$, the customer does not approve the service, and the customer evaluation is 0. The customer satisfaction function is expressed as follows:

$$f_i(t_i) = \begin{cases} 0 & 0 \le t_i \le t_{i1} \\ \dfrac{t_i - t_{i1}}{t_{i2} - t_{i1}} & t_{i1} < t_i < t_{i2} \\ 1 & t_{i2} < t_i < t_{i3} \\ \dfrac{t_{i4} - t_i}{t_{i4} - t_{i3}} & t_{i3} < t_i < t_{i4} \\ 0 & t_i \ge t_{i4} \end{cases} \tag{1}$$

The relationship between customer satisfaction and the satisfaction function of a single fresh demand point is shown in Formula (2), and the relationship between customer satisfaction function and time change is shown in Figure 1.

$$F_0 = \frac{1}{N} \sum_{i \in N} f_i(t_i) \tag{2}$$

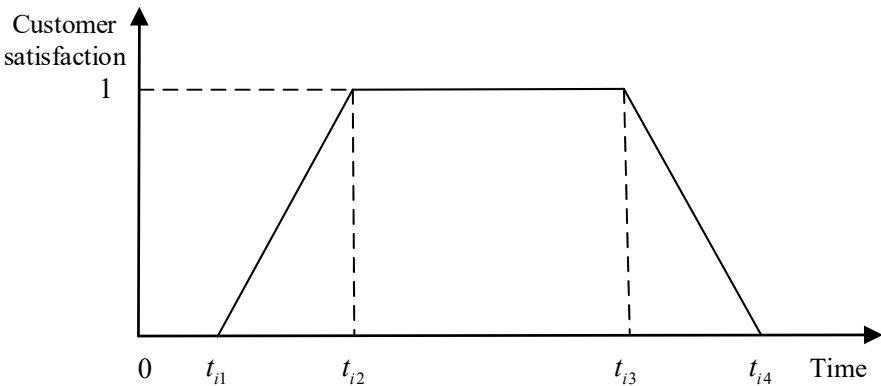

**Figure 1.** Relationship between customer satisfaction and time.

### 2.5.2. Freshness

The freshness of each perishable commodity will decrease with time. The time monotone function of continuous decline $\Omega_p(t_i)$ vis used to explain the change of freshness of fresh goods *p* with time [24]. Assuming that the freshness of fresh commodities gradually decreases after leaving the distribution center, when the departure time from the distribution center is 0, the delivery time to the customer *i* is $t_i$. The freshness of the fresh product *p* can be described as:

$$\Omega_p(t_i) = 1 - (t_i / T_p) \quad 0 \le t_i \le T_p \tag{3}$$

In Formula (3): $\Omega_p(t_i)$ represents the freshness of product p when it is in $t_i$; $T_p$ is the effective life cycle of the commodity *p*. During the delivery of goods, if the freshness of the goods is significantly reduced, this batch of goods will be rejected. Therefore, the minimum freshness condition for customers must be met within the delivery time, manifested as: $\Omega_p(t_i) \ge l_p$, as shown in Figure 2.

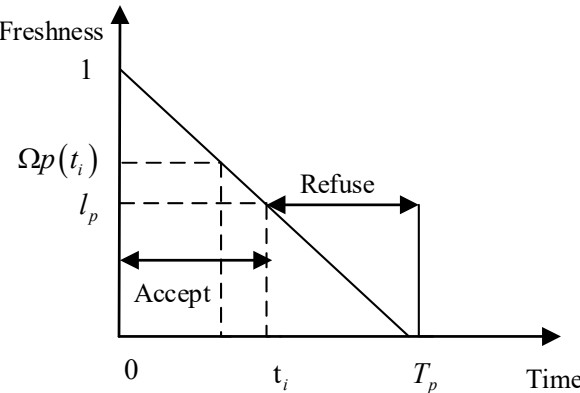

**Figure 2.** Freshness change and acceptance conditions of fresh products.

### 2.5.3. Total Distribution Cost ($F_2$) Model

(1) Fixed cost $(C_1)$ refers to the daily maintenance of distribution vehicles and the cost of employing vehicle drivers. The cost is fixed, regardless of vehicle delivery distance and load.

$$C_1 = \sum_{j \in N} \sum_{k \in K} X_{ojk} F_k \tag{4}$$

(2) Transportation cost $(C_2)$ refers to the fuel consumption cost of vehicles distributing fresh products to each demand point, which is proportional to the transportation distance:

$$C_2 = \sum_{i \in N} \sum_{j \in N} \sum_{k \in K} X_{ijk} d_{ij} C_k \tag{5}$$

(3) The refrigeration cost $(C_3)$ means that the refrigeration cost generated by the refrigeration equipment to maintain the quality of fresh products in the logistics distribution process is directly proportional to the transportation time. In this paper, the unit value of natural gas is expressed as the total refrigerant consumption in the unit period, which is as follows:

$$C_3 = \alpha\beta \sum_{i \in N} \sum_{j \in N} \sum_{k \in K} X_{ijk} t_{ij} \tag{6}$$

(4) Carbon emission $(C_4)$ cost refers to that more CO2 emissions are generated in the distribution of fresh products than in ordinary distribution. Carbon emissions are mainly generated by the combustion of vehicle fuel, where $f_c$ means the unit cost of carbon emissions and is related to the transportation distance. $\eta$ represents the fuel conversion coefficient; $a$ and $b$ are emission factors, then:

$$C_4 = f_c \sum_{i \in N} \sum_{j \in N} \sum_{k \in K} X_{ijk} \left[ \eta d_{ij} \left( a Q_{ijk} + b \right) \right] \tag{7}$$

*2.6. Model Establishment*

Objective function:

$$\max F_1 = \frac{1}{N} \sum_{i \in N} f_i(t_i) \tag{8}$$

$$\min F_2 = C_1 + C_2 + C_3 + C_4 \tag{9}$$

Constraints:

$$\sum_{i \in N} \sum_{k \in K} X_{ijk} = 1, \forall j \in N \tag{10}$$

$$\sum_{i \in N} X_{ojk} = \sum_{i \in N} X_{jok} = 1, \forall k \in K \tag{11}$$

$$\sum_{k \in K} X_{ijk} = \sum_{k \in K} X_{jik}, \forall i, j \in N \tag{12}$$

$$\sum_{p \in P} \sum_{i \in N} q_{ip} Y_{ik} \le Q_k, \forall k \in K \tag{13}$$

$$X_{ijk} = 0, \forall k \in K, i = j \in N \tag{14}$$

$$\Omega_p(t_i) = 1 - (t_i / T_p) \tag{15}$$

$$\Omega_p(t_i) \ge l_p, \forall i \in N \tag{16}$$

$$t_{ij} = d_{ij} / V, i, j \in N \tag{17}$$

$$t_j = \sum_{i \in N} \sum_{k \in K} X_{ijk} \left( t_i + \Delta t_i + t_{ij} \right), \forall j \in N \tag{18}$$

$$X_{ijk} \in \{0,1\}, \forall i, j \in N, k \in K \tag{19}$$

$$Y_{jk} \in \{0,1\}, \forall i, j \in N, k \in K \tag{20}$$

In the multi-objective vehicle routing model, the objective function formula (8) indicates the maximization of customer satisfaction, and the objective function formula (9) indicates the minimization of the total cost of fresh cold-chain distribution. Equation (10) indicates that each customer demand point is delivered once by one vehicle; Formula (11) indicates that the logistics both starts and ends in distribution centers *O*. Equation (12) indicates that vehicles arriving at the demand point must leave; Formula (13) indicates that the overload of the refrigerated vehicle is limited. Equation (14) means that there is no path between the same two points. Equation (15) refers to the freshness of the fresh product *p* when it reaches the demand point *i*. Formula (16) indicates that the freshness of fresh product *p* must meet the minimum freshness requirement when reaching the demand point *i*. Equation (17) shows the transportation time $t_{ij}$ from the demand point *i* to the demand point *j*. Equation (18) shows that the time to reach the demand point *j* is equal to the time to reach point *i* plus the service time at point *i* and the transportation time between the two points. Equations (19) and (20) are expressed in $X_{ijk}, Y_{ik}$ as 0–1 decision variables.

The basic idea when dealing with the multi-objective optimization model problem is to convert the multi-objective problem into one or a series of single-objective optimization models for an easy solution. The comprehensive objective function value $F$ takes the maximum customer satisfaction $F_1$ and the minimum total distribution cost $F_2$ as the objectives, and the corresponding formula is:

$$\min F = \frac{F_2}{F_1} \tag{21}$$

## 3. Hybrid Particle Swarm Optimization

Particle swarm optimization (PSO) is a swarm intelligence algorithm designed by simulating the predatory behavior of birds. With a fast convergence speed, it has the advantages of setting fewer parameters but with the disadvantages of a weak local search ability and a low search accuracy.

Genetic algorithm (GA) is restricted by the survival law, preference, and jigging law of naturally adapted individuals; that is, the survival law of naturally adapted individuals. Genetic algorithm has the advantage of global search, but its convergence speed is slow, and it is easy to premature. Therefore, the operation of the genetic algorithm is added to the particle swarm optimization algorithm to obtain a better solution within a shorter time. The flow of the hybrid particle swarm optimization algorithm is as follows (Figure 3).

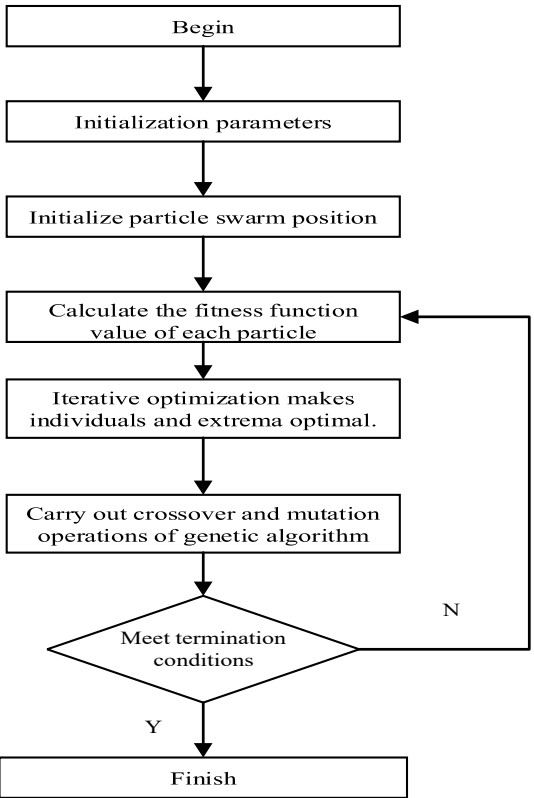

**Figure 3.** Hybrid particle swarm optimization algorithm flow.

### 3.1. Inertia Weight

The algorithm can be carried out quickly in the early stage and has excellent exploration performance. Due to the higher iteration frequency, more efficient optimization can be achieved in the later stage, and there is a large application space. The maximum linear time-varying inertia weight is introduced. The calculation coverage of the weight is $\left[\omega_{\min},\omega_{\max}\right]$, and the maximum number of iterations is $T$, expressed as:

$$\omega_t = \omega_{\max} - \frac{\left(\omega_{\max} - \omega_{\min}\right) * t}{T} \tag{22}$$

### 3.2. Introduction of Cross-Mutation Operation

The hybrid particle swarm optimization algorithm seeks the optimal solution through iteration and finds the final solution. The velocity of the particle corresponds to Formula (23), and the position of the particle corresponds to Formula (24):

$$V_i^{t+1} = \omega V_i^t + c_1\left(P_{\max} - X_i^t\right) + c_2\left(G_{\max} - X_i^t\right) \tag{23}$$

$$X_i^{t+1} = X_i^t + V_i^{t+1} \tag{24}$$

$V_i^t$ and $X_i^t$, respectively, represent the velocity and position of particle $i$ in the $t$ iteration. $c_1$, $c_2$ are self-learning factors and group learning factors. $P_{\max}$ is the best position of the particle itself and $G_{\max}$ is the whole particle group.

### 3.3. Encoding and Decoding

In the process of solving the model, this paper uses the particle swarm optimization algorithm, including crossover and mutation operations in the genetic algorithm. Because

the particle swarm optimization algorithm has good search performance in solving continuous space optimization problems, this paper adopts the real number encoding method. The speed and position of particles are n-dimensional real vectors. The integer part of each dimension of the speed represents the vehicle number and the decimal part of the same vehicle number. The size order corresponds to the distribution order of vehicles. The value of each particle position is [1, 1.999], and the speed value range is [−0.999, 0.999]. Suppose that the demand point of five fresh products distributed by the same vehicle is [1–5], and the corresponding position vector is [1.735, 1.136, 1.455, 1.892, 1.247].

### 3.4. Operation of Hybrid Particle Swarm Optimization

Step 1: Initialize parameters, import relevant data information, population size $N_L$, crossover probability $p_a$, mutation probability $p_b$, learning factor, and maximum iteration number in the cold-chain distribution of fresh products;

Step 2: Initialize particle swarm position. By initializing the velocity component and the displacement component of the particle, the single extremum of the original particle $P_{max}$ and the global extremum of the original population $G_{max}$ are determined;

Step 3: Calculate the fitness function value of each particle. In this paper, the value of the fitness function $F$ is equal to the value of the comprehensive objective function. The objective of logistics route optimization of fresh goods is to minimize the value of the comprehensive objective function $F$ because the value of the comprehensive objective function $F$ = total distribution cost $F_2$ /customer satisfaction $F_1$. Among them, the cost of expected distribution is the lowest, and customer satisfaction is maximized.

Step 4: Iterative optimization, update the velocity and position of each particle according to Equations (23) and (24) to optimize the individual and extreme value. If the particle size is equal to a single extreme value, it can be set as the extreme value of the whole system $P_{max}$. If the fitness value of the particle is equal to the global extreme value, it is set as the extreme value of the system $G_{max}$.

Step 5: Carry out crossover and mutation operations of genetic algorithm to avoid falling into local optimum.

Step 6: Check whether the termination conditions are met. If Yes, go to Step 7; or return to Step 3.

Step 7: Finish. The optimal solution is output, and the algorithm operation ends.

## 4. Example Analysis

### 4.1. Example Data

To verify the validity and correctness of the model, the following simulation examples are used [23]. A fresh company distributes fresh products to 28 customer demand points in the region through refrigerated trucks. The supply frequency is once a month. Both the starting point and the end point of the distribution are the distribution center. The refrigeration cost per unit time of the company's refrigerated truck $\alpha\beta$ = 9 RMB/h, and the average transportation cost of refrigerated vehicles is 2.1 RMB/km. The distribution center provides three types of fresh products: A, B, and C, with effective life cycles of 23, 25, and 26 h, respectively. The minimum freshness requirements of the three types of fresh products are all 75% or above. Customers have their own fuzzy time window and fresh product demand. The 2021 monthly average demand and time window requirements of 28 fresh product demand points are shown in Table 1.

**Table 1.** Information on fresh food demand points.

| Number | Position/km | A Product /kg | B Product /kg | C Product /kg | Satisfactory Time Window/h | Acceptable Time Window /h | Service Duration/h |
|---|---|---|---|---|---|---|---|
| 0 | (45,55) | - | - | - | - | - | - |
| 1 | (18,75) | 150 | 100 | 150 | [2–4] | [1–5] | 0.5 |
| 2 | (24,125) | 150 | 150 | 100 | [2.5–4.5] | [1.5–5.5] | 0.5 |
| 3 | (133,69) | 50 | 100 | 100 | [1–3] | [0–4] | 0.2 |
| 4 | (130,52) | 100 | 200 | 100 | [2–4] | [1–5] | 0.5 |
| 5 | (41,126) | 100 | 100 | 200 | [1.5–3.5] | [0.5–4.5] | 0.5 |
| 6 | (115,77) | 150 | 200 | 150 | [2–4] | [1–5] | 0.5 |
| 7 | (18,14) | 300 | 0 | 200 | [1.5–4.5] | [0.5–5.5] | 0.5 |
| 8 | (126,28) | 100 | 50 | 200 | [4–6] | [3–7] | 0.5 |
| 9 | (138,101) | 100 | 100 | 250 | [3.5–5.5] | [2.5–6.5] | 0.5 |
| 10 | (131,36) | 200 | 200 | 150 | [3–5] | [2–6] | 0.5 |
| 11 | (57,20) | 0 | 400 | 250 | [1–5] | [2–4] | 1 |
| 12 | (117,40) | 200 | 0 | 350 | [1–3] | [0–4] | 0.5 |
| 13 | (88,124) | 120 | 200 | 50 | [4–6.5] | [3–7.5] | 0.5 |
| 14 | (77,145) | 150 | 250 | 200 | [1–3] | [0–4] | 0.5 |
| 15 | (23,35) | 400 | 200 | 100 | [1.5–3.5] | [0.5–4.5] | 1 |
| 16 | (9,98) | 100 | 250 | 150 | [4–6] | [3–7] | 0.5 |
| 17 | (114,146) | 200 | 150 | 100 | [4.5–6] | [3.5–7] | 0.5 |
| 18 | (138,115) | 0 | 250 | 200 | [3–5] | [2–6] | 0.5 |
| 19 | (6,112) | 200 | 50 | 300 | [1.5–3.5] | [0.5–4.5] | 0.5 |
| 20 | (131,139) | 100 | 250 | 100 | [4–6] | [3–7] | 0.5 |
| 21 | (36,109) | 150 | 100 | 150 | [3–5] | [2–6] | 0.5 |
| 22 | (79,131) | 100 | 100 | 50 | [3.5–5.5] | [2.5–6.5] | 0.2 |
| 23 | (83,97) | 400 | 200 | 100 | [2–5] | [1–6] | 1 |
| 24 | (151,121) | 150 | 100 | 200 | [4.5–6.5] | [3.5–7.5] | 0.5 |
| 25 | (54,136) | 200 | 100 | 200 | [3.5–5.5] | [2.5–6.5] | 0.5 |
| 26 | (28,117) | 150 | 200 | 150 | [4–6] | [3–7] | 0.5 |
| 27 | (132,130) | 200 | 150 | 100 | [2–4] | [1–5] | 0.5 |
| 28 | (124,57) | 200 | 200 | 200 | [1.5–3.5] | [0.5–4.5] | 0.5 |

### 4.2. Example Solution

The programming software is Matlab2020a, and the design parameters are based on hypothetical data. The fixed cost of each vehicle $F_k$ is 70 RMB. The refrigerated vehicle travels at an average speed of 50 km/h in the distribution area. The rated load capacity of each refrigerated vehicle is 2000 kg, indicating that the unit cost of carbon emission is 0.005

RMB and the fuel conversion coefficient is 2.62; A and B are emission coefficients, where A = 0.0026 and B = 0.23. The departure time of the refrigerated truck transporting fresh food from the distribution center O is recorded as 0 h, and it arrives at each fresh food demand point in turn for distribution. The population size is 150, the maximum iteration number $T$ is 100, the crossover probability $P_a$ is 0.8, the mutation probability $P_b$ is 0.09, the individual learning factor $c_1$ is 1.5, and the global learning factor $c_2$ is 2.

By analyzing the influence of the freshness constraint on the results under the same conditions, it can be seen from Table 2 that when considering the freshness constraint, the fixed cost is reduced by 70 RMB, the transportation cost by 355 RMB, the refrigeration cost by 31 RMB, the carbon emission cost by 1 RMB, the total distribution cost by 457 RMB, and the customer satisfaction is increased by 3.9% compared with the results obtained without considering the freshness constraint. Therefore, it is necessary to consider the freshness constraint.

**Table 2.** Comparative analysis of freshness constraints.

| Constraint Condition | Fixed Cost/RMB | Transportation Cost/RMB | Refrigeration Cost/RMB | Carbon Emission Cost/RMB | Total Distribution Cost/RMB | Customer Satisfaction |
|---|---|---|---|---|---|---|
| Consider freshness (≥75%) | 490 | 3385 | 290 | 53 | 4218 | 88.22% |
| Not consider freshness | 560 | 3740 | 321 | 54 | 4675 | 84.32% |

As can be seen from Table 3, when other conditions are unchanged, the hybrid particle swarm optimization algorithm is always better than the particle swarm optimization algorithm and the genetic algorithm in the changes of different freshness constraints, which indicates the effectiveness of the multi-objective vehicle path planning model with time window and the hybrid particle swarm optimization algorithm. Moreover, it is found that the total cost of distribution and customer satisfaction is positively correlated with the freshness constraint under other conditions.

**Table 3.** Algorithm running results.

| Freshness Constraints | Algorithm | Running Time/s | Total Distribution Cost/RMB | Customer Satisfaction |
|---|---|---|---|---|
| ≥80% | HPSO | 66 | 4 627 | 90.27% |
| | PSO | 71 | 4 801 | 87.34% |
| | GA | 84 | 5 221 | 83.17% |
| ≥75% | HPSO | 62 | 4 218 | 88.22% |
| | PSO | 67 | 4 580 | 84.40% |
| | GA | 78 | 4 929 | 80.75% |
| ≥70% | HPSO | 57 | 3 230 | 80.56% |
| | PSO | 64 | 3 954 | 77.62% |
| | GA | 69 | 4 362 | 76.13% |

In addition, when the minimum freshness requirement of each fresh product is 75%, the total distribution cost obtained by the hybrid particle swarm optimization algorithm is 4218 RMB, and customer satisfaction is 88.22%. While meeting the freshness constraint conditions, the results obtained by the particle swarm optimization algorithm reduce the total distribution cost by 362 RMB, and customer satisfaction is increased by 3.82%. In other words, it has high customer satisfaction and low total distribution costs. The iterative convergence curves of hybrid particle swarm optimization and particle swarm optimization are shown in Figure 4.

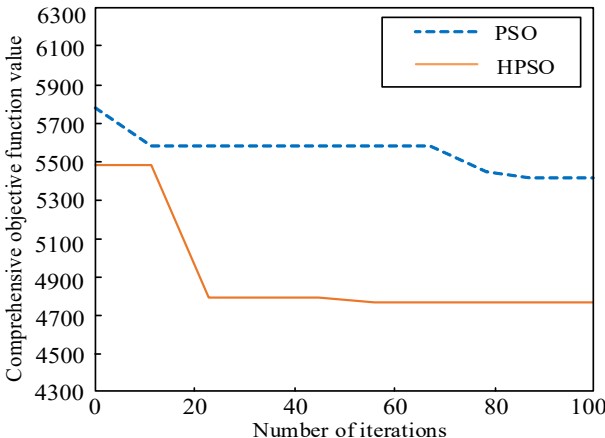

**Figure 4.** Comparison of algorithm iteration process.

The distribution line, as shown in Figures 5 and 6, suggests that the vehicle travels a total distance of 1612 km and 1740 km, respectively; thus, the hybrid particle swarm algorithm generates an optimal delivery route with less traveling than the particle swarm algorithm. Figure 4 illustrates that the optimal objective function of the hybrid particle swarm algorithm is better than that of the particle swarm algorithm. Therefore, the optimal route scheme is the hybrid particle swarm optimization algorithm. The results are shown in Table 4.

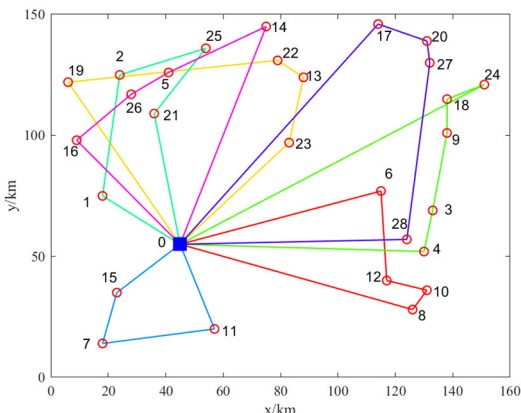

**Figure 5.** Hybrid PSO algorithm distribution path.

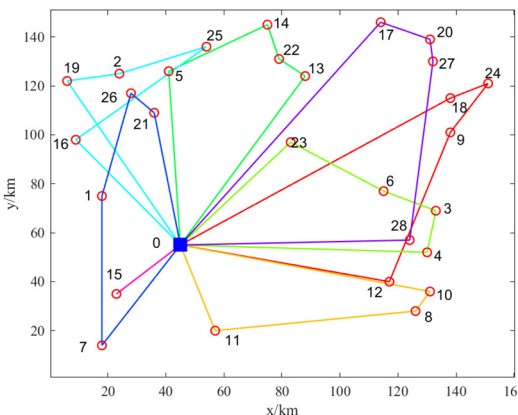

**Figure 6.** PSO algorithm distribution path.

**Table 4.** Optimal distribution scheme.

| Vehicle Serial Number | Delivery Routes |
|---|---|
| 1 | $0 \rightarrow 6 \rightarrow 12 \rightarrow 10 \rightarrow 8 \rightarrow 0$ |
| 2 | $0 \rightarrow 19 \rightarrow 22 \rightarrow 13 \rightarrow 23 \rightarrow 0$ |
| 3 | $0 \rightarrow 4 \rightarrow 3 \rightarrow 9 \rightarrow 18 \rightarrow 24 \rightarrow 0$ |
| 4 | $0 \rightarrow 1 \rightarrow 2 \rightarrow 25 \rightarrow 21 \rightarrow 0$ |
| 5 | $0 \rightarrow 15 \rightarrow 7 \rightarrow 11 \rightarrow 0$ |
| 6 | $0 \rightarrow 28 \rightarrow 27 \rightarrow 20 \rightarrow 17 \rightarrow 0$ |
| 7 | $0 \rightarrow 14 \rightarrow 5 \rightarrow 26 \rightarrow 16 \rightarrow 0$ |

*4.3. Analysis of Path Optimization Results*

In this paper, a mathematical model of MOVRPTW is established, which is solved by the hybrid particle swarm optimization algorithm considering the optimization of the double objective function. Taking part of the cold-chain logistics distribution data of a fresh product company as an example, Green techniques, including models and algorithms, are used to analyze the impact of freshness constraints on customer satisfaction and total delivery cost and compare the results of hybrid particle swarm optimization and of particle swarm optimization under different freshness constraints. It is found that customer freshness requirements have a certain impact on the total distribution cost and customer satisfaction. Enterprises make appropriate distribution path adjustments according to customer freshness requirements and balance the relationship between total distribution costs and customer satisfaction in the distribution of fresh products. Moreover, the optimal path planning scheme of multi-type fresh products under the constraint of 75% freshness is proposed.

The obtained distribution route meets the requirements of the optimal objective function value. The results obtained by the hybrid particle swarm optimization algorithm are compared with that of the particle swarm optimization algorithm and that of the genetic algorithm under the condition of no freshness constraint and different freshness constraints. The total distribution cost is the lowest, and customer satisfaction is the highest. The freshness of fresh products meeting the specified requirements means that the resource loss of fresh products can reduce the transportation cost, and the carbon emission cost can reduce the vehicle pollution emission, so it meets the requirements of the optimal objective function result of the model and algorithm.

Result display the distribution line meets the constraint conditions, and the cold-chain logistics of fresh products satisfy clients' demand for freshness. There is no overload phenomenon in the process of distribution that shows green technology. This paper concludes that the distribution of route is reasonable, not only to meet the practical demand of a cold-chain logistics company but also for the sustainable development of cold-chain logistics industry distribution playing a positive role.

## 5. Conclusions

(1) Considering fresh product freshness in fresh cold-chain distribution and customers' fuzzy time window requirements, a multi-objective green technology vehicle routing model with a time window is established. The influence of freshness constraint is analyzed. The results show that under the minimum 75% freshness constraint, the total distribution cost is reduced by 457 RMB, and customer satisfaction is increased by 3.9%. Therefore, the freshness constraint plays a positive role in the distribution of a variety of fresh products.

(2) Hybrid particle swarm optimization algorithm, particle swarm optimization algorithm, and genetic algorithm are used to analyze the total delivery cost and customer satisfaction results under different freshness constraints. The results show that the freshness constraints obtained by the hybrid particle swarm optimization algorithm are above 80%, above 75%, and above 70%, respectively. Compared with the hybrid particle swarm optimization algorithm, the total distribution cost of the other two algorithms increases by 409 RMB and 1397 RMB, and customer satisfaction increases by 2.05% and 9.71%, respectively. This shows that the green technology, including the hybrid particle swarm optimization algorithm, is superior.

(3) In summary, through the results of different freshness constraints, it can be seen that the hybrid particle swarm optimization algorithm has better results than the particle algorithm and genetic algorithm, which indicates the effectiveness of the algorithm adopted in this paper. The multi-variety fresh product distribution path obtained by the algorithm is reasonable and has a certain reference value for multi-variety raw product distribution path planning. Therefore, the results obtained by green technology in this paper are conducive to the sustainable development of the cold-chain logistics industry.

(4) At present, the limitation of this research lies in the narrow application scope of this green technology. This green technology should not only be applied in the field of cold-chain logistics distribution of fresh products but also in fields such as medical logistics and reverse logistics so as to better achieve the goal of sustainable development. The prospect of future research is to apply this green technology to logistics path planning in more scenarios so as to reduce resource consumption and vehicle pollution emissions on the premise of saving distribution costs and realizing sustainable development of the social economy and natural resources.

**Author Contributions:** Conceptualization, L.Z. and Q.L.; methodology, L.Z.; software, Q.L.; validation, Q.L.; formal analysis, L.Z.; investigation, Q.L.; resources, F.L.; data curation, F.L.; writing—original draft preparation, Q.L.; writing—review and editing, Q.L.; visualization, Q.L.; supervision, F.L.; project administration, L.Z.; funding acquisition, C.J. All authors have read and agreed to the published version of the manuscript.

**Funding:** This work is supported by the National Natural Science Foundation of China (72101082); the Youth Top Talent Project of Research Project of Humanities and Social Sciences in Colleges and Universities of Hebei Province (BJ2021088); the Natural Science Foundation of Hebei Province (F2021208011); Funding project for innovation ability training of postgraduate students in Hebei University of science and technology (XJCXZZSS2022002).

**Institutional Review Board Statement:** Not applicable.

**Informed Consent Statement:** Not applicable.

**Data Availability Statement:** The data used to support the findings of this study are included within the article.

**Conflicts of Interest:** The authors declare that there is no conflict of interest regarding the publication of this paper.

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
