# Peer review of "Research on Green Technology Path of Cold-Chain Distribution of Fresh Products Based on Sustainable Development Goals"

_sustainability, doi:10.3390/su142416659_

Round 1

Reviewer 1 Report

1. Reference should follow APA 7th edition and in-text citation should be improved. For example L Liu Y Liu should be Liu only.

2. While it is not wrong for the title to use distribution keyword but this is more of transportation research. Some scholars view logistics as part of distribution but transportation is mostly part of logistics and the model proposed by authors is related to transportation which is a logistics issue. Therefore it is recommended to use transportation instead of distribution in the title. 

3. The assumptions made in this paper is based on what? Is there any literature or theory? 

4. Example data: is this data based on real data or dummy data? Please provide citation. 

5. Example solution: the parameters such as the speed and fixed cost is this based on research or real data? Please provide citation. 

6. Analysis is based on example data. If this is based on real data you should include demographic profile of the company and place. 

7. Discussion is based on the methodology but the method with the current parameters are Insufficient for originality since the contribution is not much and the algorithm is not tested with real data. Authors should use real data with real company and location.

Author Response

Reviewer 1

Point 1:Reference should follow APA 7th edition and in-text citation should be improved. For example L Liu Y Liu should be Liu only.

Response 1:References have been revised in accordance with APA Version 7 and in-text citation has been improved.

Point 2:While it is not wrong for the title to use distribution keyword but this is more of transportation research. Some scholars view logistics as part of distribution but transportation is mostly part of logistics and the model proposed by authors is related to transportation which is a logistics issue. Therefore it is recommended to use transportation instead of distribution in the title.

Response 2:The title has been changed to “Research on Green Technology Path of cold chain Trasportation of  Fresh Products Based on Sustainable Development Goals”

Point 3:The assumptions made in this paper is based on what? Is there any literature or theory?

Response 3:Added before article assumption

In order to solve the problem from the perspective of theory and feasibility [24], The established multi-objective vehicle route planning model mainly considers the following aspects.

Point 4: Example data: is this data based on real data or dummy data? Please provide citation.

Response 4:The modified data is based on virtual simulation data and has been modified as

To verify the validity and correctness of the model, the following simulation examples are used.a fresh company distributes fresh products to 28 customer demand points in the region through refrigerated trucks.

Point 5:Example solution: the parameters such as the speed and fixed cost is this based on research or real data? Please provide citation.

Response 5:Parameters such as speed and fixed cost are based on research assumptions, Modified to

The programming software is Matlab2020a, and design parameters are based on hypothetical data.

Point 6:Analysis is based on example data. If this is based on real data you should include demographic profile of the company and place.

Response 6:

Based on the virtual simulation data, this paper studies the comparative analysis of the path planning results obtained by different algorithms under different freshness constraints.

Point 7:Discussion is based on the methodology but the method with the current parameters are Insufficient for originality since the contribution is not much and the algorithm is not tested with real data. Authors should use real data with real company and location.

Response 7:This paper mainly designs a Hybrid Particle Swarm Optimization (HPSO) algorithm to solve the minimum freshness requirement of cold chain distribution of different fresh products. The purpose of using simulation data is to verify the effectiveness of the algorithm.

Reviewer 2 Report

The article in question proposes to solve the problem of the Green Technology Path of cold chain Distribution 2 of Fresh Products Based on Sustainable Development Goals. The problem is interesting and important, however, the text has a series of observations that must be considered. 

1. The mathematical model is not tested even for small instances or used to calculate a lower bound of the solution to verify the efficiency of the genetic algorithm in small instances. I recommend that at least a relaxation of the problem is solved since it could provide some insights about the lower bounds for instances.

2. There is no detailed description of how the developed approach deals with constraint satisfaction in Figure 3. Then it is not possible to understand which approach is employed to fulfill the constraints from (10)-(18): constraints penalty in the objective function, feasibility repair function?

3. Since there are many objective functions, it should have a more detailed description of how to find the weights of each objective function.

4. It would be welcome a subsection describing in detail, with a small numerical example, the necessary encoding that connects the integer variables from the mathematical model and the individuals employed in the hybrid particle swarm optimization (HPSO).

5. There is no description of the methodology adopted to perform the calibration of parameters of the HPSO, also, experiments carried out, and results obtained with small instances. 

6. A standard repository of literature is not used for testing, nor is there a substantial set of instances to carry out tests in the developed method. This makes it impossible to perform statistical tests to obtain whether the method's performance is comparatively better than others. 

Author Response

Reviewer 2

The article in question proposes to solve the problem of the Green Technology Path of cold chain Distribution 2 of Fresh Products Based on Sustainable Development Goals. The problem is interesting and important, however, the text has a series of observations that must be considered.

Point 1:The mathematical model is not tested even for small instances or used to calculate a lower bound of the solution to verify the efficiency of the genetic algorithm in small instances. I recommend that at least a relaxation of the problem is solved since it could provide some insights about the lower bounds for instances.

Response 1:This paper mainly tests the distribution route results obtained by each algorithm under different freshness constraints, and reflects the algorithm efficiency through running time.

Point 2:There is no detailed description of how the developed approach deals with constraint satisfaction in Figure 3. Then it is not possible to understand which approach is employed to fulfill the constraints from (10)-(18): constraints penalty in the objective function, feasibility repair function?

Response 2:Figure 3 mainly describes the calculation process of the hybrid particle swarm optimization algorithm. Constraints are satisfied through the publicity in the model,

Point 3:Since there are many objective functions, it should have a more detailed description of how to find the weights of each objective function.

Response 3:The objective function of this paper mainly includes the minimum of total distribution cost and the maximum of customer satisfaction. Because the double objective function is converted into a single objective function by using , the objectives of maximum customer satisfaction and minimum total distribution cost can be achieved.

Point 4:It would be welcome a subsection describing in detail, with a small numerical example, the necessary encoding that connects the integer variables from the mathematical model and the individuals employed in the hybrid particle swarm optimization (HPSO).

Response 4:modified to

In the process of solving the model, this paper uses the particle swarm optimization algorithm including crossover and mutation operations in genetic algorithm. Because the particle swarm optimization algorithm has good search performance in solving continuous space optimization problems, this paper adopts the real number encoding method. The speed and position of particles are n-dimensional real vectors. The integer part of each dimension of the speed represents the vehicle number, and the decimal part of the same vehicle number, The size order corresponds to the distribution order of vehicles. The value of each particle position is [1,1.999], and the speed value range is [-0.999, 0.999]. Suppose that the demand point of five fresh products distributed by the same vehicle is [1,2,3,4,5], and the corresponding position vector is [1.735, 1.136, 1.455, 1.892, 1.247].

Point 5:There is no description of the methodology adopted to perform the calibration of parameters of the HPSO, also, experiments carried out, and results obtained with small instances.

Response 5:HPSO parameters refer to literature [23]. Results of experiments and small examples are shown in Table 3 and Table 4.

Point 6: A standard repository of literature is not used for testing, nor is there a substantial set of instances to carry out tests in the developed method. This makes it impossible to perform statistical tests to obtain whether the method's performance is comparatively better than others.

Response 6:The effectiveness of the algorithm is verified by the minimum constraints of different freshness.

Round 2

Reviewer 2 Report

The authors made several improvements in answering the points raised by this reviewer. Although, one point still remains: How do the constraints from (10) to (20) could be fulfilled by the developed algorithm? Does the encoding ensure a feasible solution or a penalty scheme must be applied? How this could be related to the steps described in Figure 3?

Author Response

Point 1:How do the constraints from (10) to (20) could be fulfilled by the developed algorithm? Does the encoding ensure a feasible solution or a penalty scheme must be applied? How this could be related to the steps described in Figure 3?

Response 1:The constraint satisfies the constraint conditions from (10) to (20) by writing corresponding formulas and setting corresponding parameters in Matlab software. The coding considers the optimal solution obtained under the corresponding settings. Figure 3 mainly describes the operation process of the Hybrid Particle Swarm Optimization.
